# Determinants of testing for HIV among young people in Uganda. A nested, explanatory-sequential study

Dennis Kalibbala[1,2]*, Steven Kiwuwa Mpungu[3], Bashir Ssuna[1], Wani Muzeyi[1], Happiness Mberesero[1], Fred C. Semitala[4], Anne Katahoire[3], Mari Armstrong-Hough[5], Joan N. Kalyango[1], Victor Musiime[6,7]

1 Clinical Epidemiology Unit, School of Medicine, College of Health Sciences, Makerere University, Kampala, Uganda, 2 Makerere University—Johns Hopkins University Research Collaboration, Kampala, Uganda, 3 Child Health and Development Centre, College of Health Sciences, Makerere University, Kampala, Uganda, 4 Department of Medicine, Makerere University College of Health Sciences, Kampala, Uganda, 5 Department of Epidemiology and Department of Social and Behavioral Sciences, New York University, New York, New York, United States of America, 6 Department of Paediatrics and Child Health, Makerere University, Kampala, Uganda, 7 Research Department, Joint Clinical Research Centre, Kampala, Uganda

* kalibbaladennis@gmail.com

**Data Availability Statement:** The Data are available to other researchers and as a de-identified Supporting Information file.

## Abstract

Awareness of HIV serostatus helps individuals calibrate behaviour or link to care. Globally, young people (15-24years) contribute over 30% of new HIV infections. Despite progress in enhancing access to HIV services, HIV testing among young people in Uganda is below target. We determined the prevalence and factors influencing HIV testing among young people in a peri-urban district with the highest proportion of young people. We conducted a nested explanatory sequential mixed-methods study from March to May 2019 in Wakiso district. We used stratified cluster random sampling to select 397 rural and 253 urban young people from eight parishes. We collected data using questionnaires and subsequently conducted in-depth interviews with 16 purposively selected survey participants. The prevalence of testing for HIV was 80.2%. Young people related their decisions about HIV testing to self-evaluation of their risk and perceived ability to manage the consequences of a positive result. Participants reported high levels of support for HIV testing from peers, partners, and family members. They perceived health facilities as confusing, distant, expensive, and staffed by judgmental, older health workers as barriers. They felt that mobile testing points solved some of these problems, but introduced less privacy and greater confidentiality concerns. The prevalence of HIV testing among young people in Wakiso district was low compared to the UNAIDS 2030 target but among the highest in sub-Saharan Africa. Community-based programs resolve many concerns about testing at health facilities. However, there is a need to make these programs more comfortable and private.

## Introduction

In 2018, approximately 38 million people were living with HIV/AIDS, with nearly 21 million living in eastern and southern Africa [1]. In that same year, about 1.7 million people became

**Funding:** The research reported in this publication was supported by the Fogarty International Center of the National Institutes of Health under Award Number D43 TW010037 (DK) and D43 TW009607 (DK). The content is solely the responsibility of the authors and does not necessarily represent the official views of the National Institutes of Health. The funders had no role in study design, data collection, and analysis, decision to publish, or preparation of the manuscript.

**Competing interests:** The authors have declared that no competing interests exist.

newly infected with HIV [1]. Nearly a third of these people newly living with HIV were young people aged 15–24 years. Over 3.5 million young people are living with HIV globally, of whom 63% reside in East and Southern Africa [2].

The United Nations' 95-95-95 targets to end the HIV epidemic by 2030 called for 95% of PLHIV to learn about their HIV status [3]. HIV testing saves lives by facilitating earlier status awareness, earlier treatment initiation, motivation to stay HIV-free among those who test negative, interruption of vertical mother-to-child transmission, and claims to the right to health and life [4]. However, the majority of young people are unaware of their HV status [5]. In sub-Saharan Africa, only 13% of females and 9% of male adolescents had ever tested for HIV by 2016 [6]. This is a critical age group for new HIV infections in Uganda, HIV prevalence triples from young people aged 15–19 years (1.1%) to those aged 20–24 years (3.3%) [7]. Yet, ever testing for HIV among young people is much lower in those aged 15–19 years (47.3%) compared to those aged 20-24years (84.4%) [8]. This makes them more likely to remain undiagnosed until late in the course of infection with CD4 of <200 or <350 cells/mL at diagnosis among 1.6% and 16.6% respectively [8].

To reduce HIV transmission and improve quality of life, Uganda's Ministry of Health has implemented interventions focused on enabling people to know their serostatus and linking them to prevention, care, treatment, and support services [9]. These include a mix of client and provider-initiated HIV counselling and testing delivery, and providing HIV testing services at no cost to the client. However, the uptake of HIV testing remains low among young people [7]. Little is known about the prevalence of testing and its determinants among young people. We, therefore, aimed to determine the factors associated, barriers, and facilitators influencing HIV testing among young people in Wakiso District, Uganda's most densely populated district.

## Materials and methods

### Design

We conducted a nested, sequential, explanatory mixed-methods study that combined a structured prevalence survey with in-depth interviews.

### Setting

We collected data in Wakiso district, Uganda. Wakiso encircles Kampala, the capital city of Uganda [10]. The district has a population of 1,997,418 people of which 23.6% are young people [11], and experiences a higher HIV prevalence (8.0%) than the national average (6.2%) [8]. While Wakiso includes both rural and urban settlements, most (94.3%) households are within 5km of a health facility; the district has adequate and well-distributed public and private health facilities. Public facilities include two general hospitals and 15 health centers offering free HIV testing services [11].

### Sample size

We determined sample size using the single population proportion formula with the prevalence (p) of HIV testing (70.0%) as per the study conducted by Uganda Demographic and Health Survey 2016 [7], 0.5% marginal error(d) and 95% confidence interval of certainty (alpha = 0.05). Considering a design effect of two recommended for unbiased estimates of the prevalence [12], A total of 650 respondents were included in this study.

### Participant selection

We randomly selected one out of nine urban sub-counties (Nabweru, N = 250,755) and one out of the six rural sub-counties (Namayumba, N = 33,320) [10]. We used proportionate

stratified random sampling to select three of six parishes in Nabweru [Kazo(N = 33,424), Maganjo(N = 45,847), and Nansana (N = 52,107)] and five of ten parishes in Namayumba [Kitayita (N = 3,402), Kyampisi (N = 1,963), Kyanuuna (N = 4,960), Luguzi (N = 6,144), and Lutisi (N = 2,077)] [13, 14]. From each parish, we randomly selected six villages and then 14 households from each village with an eligible young person aged 15–24 years residing in the selected household. When we found a household with more than one eligible participant, we randomly selected one participant and obtained informed consent.

From the survey participants, we purposively selected 16 participants for in-depth interviews based on their HIV testing status, age, gender, and marital status.

## Data collection

We selected participants from March to May 2019. We collected data during school holidays, when young people enrolled in school typically return home. We used interviewer-administered questionnaires to collect quantitative data and interview guides to collect qualitative data. The interview guides were designed using the Capability, Opportunity, Motivation-Behavior Model (COM-B) [15]. All tools were bilingual. Interviews took place in English or Luganda (the most widely spoken language), depending on participant preference. Authors DK and HM trained ten research assistants to collect quantitative data and introduced them to local village health team members (VHTs), who guided the team to local council leaders and in the enumeration areas.

In-depth interviews were conducted in English or Luganda by two bilingual research assistants, with training and experience in conducting qualitative interviews. Interviews were conducted at the nearest public health facility for 20 to 35 minutes and were audio-recorded. Participants were interviewed about their understanding, interpretation, and experiences regarding HIV testing.

## Quality control

The questionnaire was pre-tested and the research assistants who administered it were trained and supervised during data collection. Authors DK and HM reviewed all questionnaires for completeness daily before storage. Double data entry was used to ensure quality.

## Analysis

Data were entered into EPI-DATA 4.4 software and then exported to STATA-15 for cleaning and analysis. Descriptive characteristics were calculated as frequencies and percentages. The proportion of those who had ever tested was calculated with its 95% confidence intervals after adjusting for clustering at the sub-county and parish levels. We used modified Poisson regression to adjust for survey sampling and sample weights and report clustered robust standard errors because the outcome was not rare (>20%). Factors with a p-value <0.2 at bivariate analysis were considered for the multivariate analysis. Statistical significance was determined at a p-value <0.05.

Recorded interviews were transcribed and reviewed repeatedly alongside recordings to ensure that the content was transcribed verbatim. Luganda transcripts were subsequently translated into English. Transcriptions were imported into Atlas. ti 8 and applied open coding to inductively generate the initial set of codes. Codes were then iteratively reviewed and revised with BS, WM, AK, and MAH. The revised codes were applied to the rest of the data. The codes were grouped into categories and themes were identified. DK synthesized the emergent themes and selected illustrative quotations for each theme.

## Ethics

We obtained approval from the School of Medicine Research and Ethics Committee (SOM-REC) of Makerere University (#REC REF 2019–052). We also obtained administrative clearance from the Wakiso District Health Officer, the Nansana Municipal Health Officer, local council leaders, and facility in-charges. Participants who were aged 18–24 years, and emancipated minors (individuals < 18 years who are pregnant, married, have a child, or cater for their livelihood) as categorized by national guidelines [16] individually gave informed written consent before participation. For participants < 18 years, we sought consent and assent from the guardian and the participants, respectively, before participation. For an illiterate participant, an impartial witness was invited to witness the consent. The consent was documented by providing a signature or thumbprint on the consent form after exchanging information between the researcher and research participants during the whole research process.

Participants were assigned unique identification numbers for confidentiality purposes and these numbers were maintained throughout the study.

## Results

### Participant characteristics

A total of 650 young people participated in the study. Their average age was 19(±2.6) years and the majority were female (60.9%) (Table 1).

### Prevalence of HIV testing

The prevalence of ever testing was 80.2% (95%CI: 76.9–83.1%) while the prevalence of testing in the last year was 75.0% (95%CI: 71.1–78.6%). Among those who had ever tested, self-reported HIV positive status was at 5.2% (n = 26) but 16 did not mention their status. They listed their reasons for testing and not testing (Table 2). Ever testing for HIV was significantly different between the female (83.6%) compared to the males (74.8%) (p-value = 0.006).

From the regression analysis, participants who were female (Adjusted prevalence ratio [aPR] = 1.09, 95%CI: 1.01–1.18), older(20–24 years)(aPR = 1.26, 95%CI:1.15–1.37), ever married (aPR = 1.07, 95%CI:1.01–1.14), ever had sexual intercourse(aPR = 1.13, 95%CI: 1.02–1.26), encouraged to test by peers(aPR = 1.18, 95%CI: 1.08–1.28) and aware of youth-friendly services(aPR = 1.12, 95%CI: 1.01–1.23) were more likely to test for HIV while those living >10 km from a facility(aPR = 0.77, 95%CI: 0.59–0.99) were less likely to test for HIV after adjusting for alcohol consumption (Table 3).

### Interviews

**"Why should I test?" Motivations for HIV testing.** Young people reported making decisions about HIV testing based on their perceived risk of HIV infection. Individuals who abstained from sex or lacked a current sexual partner, those who did not feel sick or have symptoms, and those who believed their parents' HIV status to be negative, perceived themselves to be free of risk and therefore not in need of testing. One male participant described testing for HIV as a woman's responsibility, not his. Although sexually active, he believed he had no reason to test. Finally, some young people perceived themselves to be at low risk of infection because they trusted condoms. Trusting that their lack of symptoms, demands on sexual partners to test, and/or use of condoms resulted in low risk, these participants were unmotivated to test for HIV (Table 4).

**Fears related to testing.** Some young people explained that they feared retesting for HIV when they perceived themselves to be at elevated risk. They felt that inconsistent use of

**Table 1. Characteristics of 650 young people in Wakiso district.**

| Characteristics | Total n(%) | Males(n = 254) n(%) | Females(n = 396) n(%) | Chi-square p-value |
|---|---|---|---|---|
| **Age (years)** | | | | 0.364 |
| 15–19 | 308(47.4) | 126(49.5) | 182(46.0) | |
| 20–24 | 342(52.6) | 128(50.5) | 214(54.0) | |
| **Marital status** | | | | <0.001* |
| Single | 493(75.9) | 214(84.3) | 279(70.5) | |
| Married | 153(23.5) | 39(14.3) | 114(28.8) | |
| Separated/Widowed | 4(0.6) | 1(0.4) | 3(0.8) | |
| **Education level** | | | | 0.237 |
| None | 13(2.0) | 6(2.4) | 7(1.8) | |
| Primary | 174(26.8) | 77(30.3) | 97(24.5) | |
| Secondary | 425(65.3) | 160(63.0) | 265(66.9) | |
| Tertiary | 38(5.9) | 11(4.3) | 27(6.8) | |
| **Employment status** | | | | <0.001* |
| Unemployed | 234(36.0) | 48(18.9) | 186(47.0) | |
| Student | 143(22.0) | 52(20.5) | 91(23.0) | |
| Formally employed | 134(20.6) | 75(29.5) | 59(14.9) | |
| Self-employed | 139(21.4) | 79(31.1) | 60(15.1) | |
| **Residence** | | | | 0.104 |
| Rural | 397(61.1) | 165(65.0) | 232(58.6) | |
| Urban | 253(38.9) | 89(35.0) | 164(41.4) | |
| **Distance to the nearest HIV testing site** | | | | 0.185 |
| <5km | 522(80.3) | 195(76.8) | 327(82.6) | |
| 5-10km | 94(14.5) | 44(17.3) | 50(12.6) | |
| >10 km | 34(5.2) | 15(5.9) | 19(4.8) | |
| **Alcohol consumption** | | | | 0.038* |
| Never | 537(82.6) | 198(78.0) | 339(85.6) | |
| Past use | 55(8.5) | 26(10.2) | 29(7.3) | |
| Current use | 58(8.9) | 30(11.8) | 28(7.1) | |
| **History of TB treatment** | | | | 0.154 |
| No | 617(94.9) | 245(96.5) | 372(93.9) | |
| Yes | 33(5.1) | 9(3.5) | 24(6.1) | |
| **Perceived risk of getting HIV/AIDS** | | | | 0.127 |
| Yes | 127(19.8) | 59(23.6) | 68(17.3) | |
| Sometimes | 57(8.9) | 23(9.2) | 34(8.7) | |
| Not really | 466(71.4) | 168(67.2) | 291(74.0) | |
| **Ever had sexual intercourse** | | | | 0.550 |
| Yes | 426(65.5) | 170(66.9) | 256(64.7) | |
| No | 224(34.5) | 84(33.1) | 140(35.3) | |
| **Condom use (n = 426)** | | | | <0.001* |
| Never | 131(30.8) | 46(18.1) | 92(23.2) | |
| Sometimes | 213(50.0) | 82(48.2) | 131(51.2) | |
| Always | 82(19.3) | 52(30.6) | 30(11.7) | |
| **Current sexual partners (n = 426)** | | | | <0.001* |
| None | 27(6.3) | 7(4.1) | 20(7.8) | |
| Only 1 | 252(59.2) | 81(47.7) | 171(66.8) | |
| 2–5 | 125(29.3) | 64(37.7) | 61(23.8) | |
| >5 | 22(5.2) | 18(10.5) | 4(1.6) | |

*(Continued)*

**Table 1.** (Continued)

| Characteristics | Total n(%) | Males(n = 254) n(%) | Females(n = 396) n(%) | Chi-square p-value |
|---|---|---|---|---|
| **Comprehensive HIV/AIDS knowledge** | | | | 0.089 |
| No | 298(45.9) | 127(50.0) | 171(43.2) | |
| Yes | 352(54.2) | 127(50.0) | 225(56.8) | |
| **Willingness to test for HIV** | | | | 0.559 |
| Yes | 623(95.9) | 242(95.3) | 381(96.2) | |
| No | 27(4.1) | 12(4.7) | 15(3.8) | |
| **Encouraged to test by peers** | | | | 0.348 |
| Yes | 406(62.5) | 153(60.2) | 253(63.9) | |
| No | 244(37.5) | 101(39.8) | 143(36.1) | |
| **Perceived HIV testing services as Youth-friendly[β] (n = 639)#** | | | | 0.155 |
| Yes | 533(83.4) | 202(80.8) | 331(85.1) | |
| No | 106 (16.6) | 48(19.2) | 58(14.9) | |

*Statistically significant

#Eleven participants did not know.

[β] Equitable, accessible, acceptable, appropriate, and effective services.

condoms, having multiple sexual partners, and engaging in compensated sex for money put them at a higher risk of HIV. Yet when they engaged in activities they perceived as risky, their motivation to test decreased because they feared being told that they were living with HIV. Others, who perceived themselves to be safe, also said they refrained from testing because they feared the results. Young people are also worried about their ability to manage the consequences of a positive result. They perceived the consequences of a positive result to be stress, growing thin, stigma, and swallowing big tablets. Finally, participants said they feared discomfort during the blood draw and loss of confidentiality at mobile testing points in the community and in both private and public facilities (Table 4B). These concerns also affected the willingness to test for HIV.

**Engagement with other health services facilitates testing.** Young people mentioned being offered HIV testing when they visit health facilities for other medical services, such as

**Table 2. Reasons for testing and not testing for HIV among young people among the 650 participants in the survey.**

| Testing (N = 521) | Not Testing (N = 129) |
|---|---|
| Desire to know the status (n = 394) | Never having had sexual intercourse (n = 29) |
| Antenatal checkup (n = 28) | Lack of time (n = 21) |
| At school programs (n = 18) | Fear of a positive result (n = 20) |
| Influenced by Peers (n = 11) | "Still young" (n = 4) |
| Elevated concern after unsafe sex (n = 10) | No support/permission from parents (n = 5) |
| Because of parental influence (n = 8) | Fear of the pain during blood draws (n = 5) |
| Had gone for voluntary medical male circumcision (n = 7) | Never been approached/advised (n = 2) |
| Had gone for family planning (n = 4) | Few health workers at the test(n = 1) |
| Accessed Government hospital (n = 4) | Had never taken it serious/ important (n = 2) |
| Fear of family background (n = 4) | Not involved so much in sex (n = 1) |
| Community outreach (n = 4) | Do not trust local testing services(n = 1) |
| Regular checkup (n = 3) | Fear to get it from facilities (n = 1) |
| To donate blood (n = 2) | No reason (n = 32) |

**Table 3. Factors associated with testing for HIV among 650 young people in Wakiso district.**

| Characteristics | Ever tested | | Unadjusted prevalence ratio(PR) (95%CI) | Adjusted PR (95%CI) |
|---|---|---|---|---|
| | No (n = 129) | Yes (n = 521) | | |
| **Sex** | | | | |
| Male | 64(25.2) | 190(74.8) | 1 | 1 |
| Female | 65(16.4) | 331(83.6) | 1.12 (1.03–1.21) | 1.09 (1.01–1.18) |
| **Age** | | | | |
| 15–19 years | 102(33.1) | 206(66.9) | 1 | 1 |
| 20–24 years | 27(7.9) | 315(92.1) | 1.38 (1.26–1.49) | 1.26 (1.15–1.37) |
| **Marital status** | | | | |
| Single | 122(24.8) | 371(75.2) | 1 | 1 |
| Married/ Widowed | 7(4.5) | 150(95.5) | 1.26(1.18–1.34) | 1.07 (1.01–1.14) |
| **Education level** | | | | |
| None/ Primary | 36(19.3) | 151(80.7) | 1 | |
| Secondary/ Tertiary | 93(20.1) | 370(79.9) | 0.99 (0.91–1.08) | - |
| **Employment status** | | | | |
| Unemployed | 39(16.7) | 195(83.3) | 1 | |
| Student | 49(34.3) | 94(65.7) | 0.79 (0.69–0.89) | - |
| Employed | 41(15.0) | 232(85.0) | 1.02 (0.95–1.10) | |
| **Distance to nearest HIV testing site** | | | | |
| <5km | 101(19.4) | 421(80.6) | 1 | 1 |
| 5-10km | 15(16.0) | 79(84.0) | 1.04 (0.94–1.15) | 1.06 (0.97–1.16) |
| >10 km | 13(38.2) | 21(61.8) | 0.77 (0.59–1.01) | 0.77 (0.59–0.99) |
| **Alcohol** | | | | |
| Never | 118(22.0) | 419(78.0) | 1 | 1 |
| Ever used | 5(9.1) | 50(90.9) | 1.16 (1.07–1.24) | 1.05 (0.97–1.13) |
| **Ever had sexual Intercourse** | | | | |
| No | 75(33.5) | 149(66.5) | 1 | 1 |
| Yes | 54(12.7) | 372(87.3) | 1.31 (1.18–1.45) | 1.13 (1.01–1.26) |
| **Condom use** | | | | |
| Never | 26(19.9) | 105(81.1) | 1 | |
| Sometimes | 17(8.0) | 196(92.0) | 1.15 (1.04–1.26) | - |
| Always | 11(13.4) | 71(86.6) | 1.08 (0.95–1.22) | |
| **Current sexual partners** | | | | |
| None | 4(14.8) | 23(85.2) | 1 | |
| Only 1 | 28(11.1) | 224(88.9) | 1.04 (0.89–1.23) | - |
| ≥2 | 22(14.9) | 125(85.1) | 0.99 (0.84–1.18) | |
| **Comprehensive HIV/AIDS knowledge** | | | | |
| No | 63(21.1) | 235(78.9) | 1 | |
| Yes | 66(18.8) | 286(81.2) | 1.03 (0.95–1.11) | - |
| **Encouraged to test by Peers** | | | | |
| No | 68(27.9) | 176(72.1) | 1 | 1 |
| Yes | 61(15.0) | 345(85.0) | 1.19 (1.08–1.28) | 1.18 (1.09–1.28) |
| **Perceived HIV testing services as Youth-friendly** | | | | |
| No | 68(27.9) | 176(72.1) | 1 | 1 |
| Yes | 61(15.0) | 345(85.0) | 1.13 (0.99–1.28) | 1.12 (1.01–1.25) |

**Table 4. Barriers and facilitators influencing HIV testing among young people in Wakiso.**

| | Sample excerpts |
|---|---|
| **A. Motivations for HIV testing** | *"Why should I test? Am I sick? Maybe if I got it at birth. I may have all other diseases but not AIDS!"* **(Id = 400, male, never tested, 19)**<br>*"I know I don't have it [HIV]. If she [sexual partner] wants to prove that I don't have it's up to her. Before I sleep with any woman, I get her results. Unless she fakes them, but she first tests before I can sleep with her."* **(Id = 289, a male, never tested, 22)**<br>*"They have tried to put condom boxes everywhere that after 3–4 houses you find a box. For me, I trust condoms. If I am using a condom, why should I test for HIV?"* **(Id = 98, male, never tested, 23)** |
| **B. Fears related to testing** | *"Men are liars. . . I just shifted to this area but previously I used to test for HIV quarterly, I now have 3 sexual partners. I always ask them to use condoms but they remove them during sex, I realize it after. . . now I fear to test again. . ."* **(Id = 571, female, ever tested, 23)**<br>*"I gave birth 2 years ago but I have never tested for HIV again because I fear to be told, I am positive."* **(Id = 706, Female, ever tested, married, 20)**<br>*"You see, they can tell that you are positive and you end up stressed/ worried. So I never tested. I fear to know. . . and fear taking drugs. . . I have a friend who was going abroad for work, but when she got tested and found that she has the infection, she wasn't taken and she got so stressed. Right now, she is thin."* **(Id = 704, a female, never tested, 19)**<br>*". . . Some may not feel comfortable thinking that they may be positive and the information is spread. . . Yes, some health workers are rumour mongers. They check you and inform someone about your results"* **(id = 500, female, ever tested, 15)**<br>*"If it was this little pricking it would be fine. . .No one doesn't fear injections even you [interviewer] [Laughs]. You cannot fear an injection on the buttocks and like the one for the arms. . ."* **(ID = 615, Male, never tested, 15)** |
| **C. Engagement with other health services facilitates testing.** | *"I first tested at 18 years and it was because I was pregnant. But I used to fear being told I was positive, so I never tested until antenatal. . . I gave birth 2 years ago but I have never tested for HIV again because I fear being told I am positive."* **(Id = 507, Ever tested, married, 21)**<br>*"Well, I was sick of malaria and they checked for everything, they checked malaria, typhoid, and HIV. There is no reason as to why I tested. . ."* **(Id = 500, female, ever tested, 15)**<br>*"From the clinics (private), you go and pay 5000 and they test you for HIV. . . while at the hospital (public) it for free. . . most cases it is lack of money that makes us not test for HIV but also sometimes there are many people where it is for free and the nurses be tough."* **(Id = 532, male, ever tested, 17).**<br>*". . .. Some health workers are rude. If the person gets tested, come talking about how rude the health workers were. Shouting at you, shaming you in front of other patients."* **(Id = 701, female, never tested, 16).**<br>*"I don't have money. From here to XX health centre, it costs us 2000ugx shillings (~0.6USD), so every time you go you must spend. . . I had not gone to test for it (HIV)."* **(Id = 224, Male, never tested, 21)** |
| **D. Mixed feelings regarding mobile testing outreaches** | *"The individuals [community] come one at a time, but there is a time they come so many yet the tent is small. They sit and wait for others to stand without being helped. Some [youth] don't want to be seen. . .. The health worker should be many enough."* **(Id = 224, a male, never tested, 21)**<br>*"No, they don't attract youth, because it's for the public and there are usually many people. Youth prefer to go test somewhere they are unknown. They don't want to be known that they have gone for HIV testing. . .They advertise and many people come, but the youth fear and don't come."* **(Id = 532, male, ever tested, 17).** |

*(Continued)*

**Table 4.** (Continued)

| | Sample excerpts |
|---|---|
| **E. Influence from peers, partners, and family members** | *"For me, my friends just advised me to protect myself. If I can also go and test it is better. They also advise my siblings that way and if they cannot control themselves they can use a condom."* **(Id = 071, Ever tested, Single, 15).** <br> *"My Aunt told me that "Mululu gwabindanzi" ["love for fried bread" to mean "sexual desire"] got her pregnant and HIV positive. She advised me to always test for HIV before sex because some people can be born with HIV. . .and can be looking fine"* **(Id = 704, female, never tested, 19).** |

when they suspect typhoid or malaria or during antenatal care (ANC). Some mentioned being first tested during ANC. First-time testing at ANC implies that testing was not routine for these individuals. Testing for HIV was also mentioned to be offered together with other medical services when someone is sick. However, young people also described feeling disengaged from health services and alienated by health facilities. Those who described receiving free HIV testing services at the public facility also said they experienced those health facilities as confusing, distant, expensive, and staffed by judgmental older health workers. Others who refrained from testing perceived health workers at public facilities to be unwelcoming, yet those attending private facilities were perceived to be expensive. Even the indirect financial costs of accessing "free" testing services at public facilities could be high due to long distances and transport costs (Table 4C).

**Mixed feelings regarding mobile testing outreaches.** Young people reported that HIV testing outreaches enabled them to test for HIV because they resolved the problems of confusing, distant, and expensive health facilities. Community outreaches were reported to provide free testing services closer to the communities. However, young people also noted that these outreaches also introduced a lack of privacy. Young people said they feared loss of confidentiality because the entire community attended these outreaches. Young people also mentioned being tested at school, while others mentioned not testing because they were too busy with school. One participant suggested that HIV testing should be made compulsory at school (Table 4D).

**Influence from peers, partners, and family members.** Participants reported that receiving support from their peers, partners, or family members regarding HIV testing facilitated their testing for HIV. Even some who have never tested described encouragement to test from those close to them. Even in the context of a supportive social environment, though, some participants said they were too busy to test (Table 4E).

## Discussion

In this study, we identified factors associated with testing for HIV among young people in a peri-urban district with the highest proportion of young people in Uganda, as well as the barriers and facilitators to HIV testing these young people perceive. Eight in ten of the young people in Wakiso had tested for HIV in their lifetime. This lifetime prevalence of HIV testing is among the highest in sub-Saharan Africa and is likely due to easy access and free testing services. The high prevalence can also be attributable to the difference in time and the fact that HIV testing varies considerably across different settings likely due to interventions and testing behaviour [17–19]. Similar studies have found the testing prevalence to be 59.3% in Kenya [20] and only 29% in Tanzania [21], though these studies included a wider age group of 13–24 years. A similar study among young people (18–24 years) in South Africa also reported a lower

prevalence of testing (52%) [22]. Nonetheless, Uganda's testing prevalence remains below the UNAIDS target of 95% [3].

In our study, young women were more likely to test than young men. This is consistent with a large body of literature finding that women have greater healthcare-seeking behaviour than men [23]. Similar findings were reported from South Africa [22] and four other sub-Saharan countries [24]. One reason for this may be that women are likely to be offered HIV testing during ANC; in in-depth interviews, young people said they are tested when they visit health facilities for other services, including ANC [25]. HIV testing programs should emphasize antenatal attendance while discouraging home births.

Young people aged 20–24 years were more likely to test for HIV than those aged 15–19 years. Older youth may be more knowledgeable about HIV/AIDS [26], more likely to have married and tested with their partners, and more likely to have become pregnant and tested during ANC [7, 22, 27].

From the interviews, decisions related to HIV testing are based on self-evaluations of the risk of HIV infection and personal capacity to manage the consequences of a positive result. Indeed, participants who had married or engaged in sexual intercourse were more likely to have been tested compared to their counterparts. This may be because they perceived themselves to be at risk of HIV or had more opportunities to do so. Previous studies found that many young people test for the first time upon marriage [27] and having at least two lifetime sexual partners is associated with HIV testing [19].

Young people who lived >10km from the nearest HIV testing facility were less likely to have tested than those who lived closer to it (<5km). Although Wakiso has many health facilities, distance remains a barrier to HIV testing for some young people. Previous studies have also found that distance restricts young people from accessing health care services including HIV testing services [21, 28]. In our study, qualitative interviews indicated that participants felt community outreaches resolved many barriers to testing, such as cost and distance. However, they emphasized that outreach testing introduces new barriers, such as a lack of privacy. Youths do not want it known that they have gone for HIV testing.

Surprisingly, we did not detect a relationship between rural or urban status and testing. Opportunities for HIV testing may be equally available among both rural and urban youths in Wakiso. In contrast, others have reported that youth in urban areas are more willing to test for HIV compared to youth in rural areas [29].

Young people who were encouraged by their peers to test for HIV were more likely to have tested compared to those whose peers did not encourage testing. This implies young people are motivated to test for HIV by their peers. Peers may be particularly effective for encouraging people from stigmatized populations, who mistrust healthcare providers [30]. In interviews, young people also mentioned encouragement and support from partners and family members. Engaging parents, family, and peers may improve the uptake of HIV testing among young people. Interventions to provide parents and peers with more and correct information, such as through 'straight talk' programs, seminars, drama, and the provision of information, education and communication materials should be considered.

Young people who perceived HIV testing services as youth-friendly were more likely to have tested for HIV compared to their counterparts. In interviews, young people described health facilities as confusing, distant, expensive, and staffed by judgmental older health workers. Peer health workers can motivate, reduce mistrust in healthcare providers, and encourage young people to test for HIV [30]. The mere presence of health facilities is insufficient; there is a need to ensure facilities are also youth-friendly.

This study has some limitations. First, HIV testing was measured by self-report. This may result in over-reporting of testing. Second, social desirability bias may have influenced

responses. Third, the survey component of this study did not study all potential factors associated with testing for HIV, such as involvement in commercial sex and drug use. However, we were able to probe respondents during the in-depth interviews about the contributors to the decision to test or not to test. Lastly, though we collected data during the holiday season, weekends, and evening hours, we were unable to capture relatively equal proportions of young people in school and those out of school.

Our study also has several strengths. First, we studied both urban and rural settings with a sufficient sample size of 650 and 84% power to detect a meaningful difference between those aged 15–19 years and 20–24 years. Second, we sequentially employed both quantitative and qualitative methods to gain a better understanding of these factors. Lastly, both quantitative and qualitative data were collected by fellow young people as research assistants trained on the protocol and how to collect data, which may have motivated participants to freely and frankly share their experiences.

## Conclusions

The prevalence of HIV testing among young people in the Wakiso district is close to the UNAIDS 2030 target. Testing more frequently is needed to meet 95-95-95 targets since many of those ever tested are not aware of their status. Married women living near a testing site, those who had peer support, and those who had ever had sexual intercourse were more likely to test for HIV. Community testing programs were preferred for health facilities. However, there is a need to make these services more comfortable and private. This could include targeted community interventions to reach more young men living far from HIV testing sites and organizing outreaches at times young people are likely to be available and at appropriate venues that make young people feel safe enough to test. Finally, many young people who had previously tested for HIV were nonetheless uncertain about their HIV status and feared retesting. Further studies should investigate HIV status awareness among young people who have previously tested for HIV.

## Supporting information

**S1 File. Interview guide.**
(DOCX)

**S1 Data. De-identified data.**
(DTA)

## Acknowledgments

We thank the Makerere University School of Medicine Implementation Science (MAK-ImS) Training program, and the Uganda Pulmonary Complications of AIDS Research Training Program (PART) at Makerere College of Health Science (MakCHS) Mixed Methods Fellowship for the various training and technical support offered to this study. We also thank all VHTs and Research Assistants who conducted field data collection, and the respondents for participating in the study.

## Author Contributions

**Conceptualization:** Dennis Kalibbala, Steven Kiwuwa Mpungu, Fred C. Semitala, Joan N. Kalyango, Victor Musiime.

**Formal analysis:** Dennis Kalibbala, Mari Armstrong-Hough.

**Investigation:** Dennis Kalibbala.

**Methodology:** Dennis Kalibbala, Bashir Ssuna, Wani Muzeyi, Fred C. Semitala, Anne Kata-
hoire, Mari Armstrong-Hough, Victor Musiime.

**Supervision:** Steven Kiwuwa Mpungu, Fred C. Semitala, Joan N. Kalyango, Victor Musiime.

**Writing – original draft:** Dennis Kalibbala, Wani Muzeyi, Happiness Mberesero, Fred C.
Semitala, Anne Katahoire, Mari Armstrong-Hough, Joan N. Kalyango, Victor Musiime.

**Writing – review & editing:** Steven Kiwuwa Mpungu, Bashir Ssuna, Wani Muzeyi, Happiness
Mberesero, Fred C. Semitala, Anne Katahoire, Mari Armstrong-Hough, Joan N. Kalyango,
Victor Musiime.

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
