## [Decision Letter · Decision Letter 0]

22 Nov 2021

PGPH-D-21-00782

Determinants of testing for HIV among young people in Uganda. A nested, explanatory-sequential study

Dear Dr. Kalibbala,

Thank you for submitting your manuscript to PLOS Global Public Health. After careful consideration, we feel that it has merit but does not fully meet PLOS Global Public Health’s publication criteria as it currently stands. Therefore, we invite you to submit a revised version of the manuscript that addresses the points raised during the review process.

We look forward to receiving your revised manuscript.

Kind regards,

Kwabena Obeng Duedu, PhD

Academic Editor

Journal Requirements:

1. Please provide additional details regarding participant consent. In the ethics statement in the Methods and online submission information, please ensure that you have specified what type of consent you obtained (for instance, written or verbal, and if verbal, how it was documented and witnessed). 

If your study included minors, state whether you obtained consent from parents or guardians.

2. In the online submission form, you indicated that "The datasets used and/or analyzed during the current study are available from the corresponding author on reasonable request."

3. State the initials, alongside each funding source, of each author to receive each grant.

Additional Editor Comments (if provided):

Your paper was of interest to the reviewers and falls within the scope of interest of PGPH. As you can see below, the reviewers identified both strengths and weaknesses in your study. The reviewers noted important areas of your paper that require careful attention. From my own reading of your paper, I agree with the reviewers in that I believe that your paper could make an important contribution and that your paper will benefit from a major revision.

Although the limitations of your current paper are important for you to attend to, I am also in agreement with the reviewers that even the most serious limitations appear amenable to revision. I am therefore unable to accept your paper for publication in PLOS Global Public Health and I am inviting you to revise and resubmit your paper to the journal.

Reviewers' comments:

Reviewer's Responses to Questions

**Comments to the Author**

1. Does this manuscript meet PLOS Global Public Health’s publication criteria? Is the manuscript technically sound, and do the data support the conclusions? The manuscript must describe methodologically and ethically rigorous research with conclusions that are appropriately drawn based on the data presented.

Reviewer #1: Yes

Reviewer #2: Yes

2. Has the statistical analysis been performed appropriately and rigorously?

Reviewer #1: Yes

Reviewer #2: Yes

3. Have the authors made all data underlying the findings in their manuscript fully available (please refer to the Data Availability Statement at the start of the manuscript PDF file)?

Reviewer #1: Yes

Reviewer #2: Yes

4. Is the manuscript presented in an intelligible fashion and written in standard English?

Reviewer #1: Yes

Reviewer #2: Yes

5. Review Comments to the Author

Reviewer #1: 1. Some of the study participants reported history of ever HIV testing and more than 5% of them were positive. It is essential to discuss the influence of local HIV infection prevalence on the observed results.

2. In Table 1, the characteristics of the study participants should be displayed in subgroups according to rural/urban status according to the used stratified cluster random sampling method. I would suggest to list rural/urban status in this table 3 as well

3. In Table 3, there is a wrong number in "Ever had sexual Intercourse" as "149(87.3)".

4. The identified independent determiants in table 3 mostly showed associations in relatively weak strength. Please discuss the significance of such findings for directing practice.

Reviewer #2: Any more info on

a) gender, study focuses on these two age groups, but would have provided insight to breakdown by gender as well. I would at least bring findings more into the discussion as likely there are more differences. would show if age, marital status, ever had sex vary by gender. Gender disaggregated data would likely be insightful.

b) locality prevalence rates (again by gender would be helpful)

b) Is there any more insight into knowledge of treatment options, effectiveness of treatment? often helps improve test-seeking

c) Any insight into % of people who know someone close to them with HIV, often helps improve testi-seeking

d) ANC testing rates? if facility based birth, all actually tested? % facility based? % who have a child or are pregnant/have pregnant partner?

e) Healthcare interactions regularly lead to testing or offers for testing, especially hospitalization and ANC, but also other clinic interactions? is this the norm?

Line 102. approx size of parishes. how many residents or households?

Line 108 gender not used in selection?

Line 178 distance (lower case)

Line 183 Is Table 4 necessary? Given the population of Namayumba is under 20K this would be consider Protected Health Information in general. It is not clear this adds to the document.would rather just a brief description of ages, geography, gender etc

Line 205 Any information on perceived availability of treatment? know people who are treated? know that treatment is effective. Any impact of knowing someone with HIV?

Line 250 Isn't the 95-95-95 that 95% of PLHIV know their diagnosis. It's slightly different than "testing prevalence remains below the UNAIDS target of 95%"

Line 317 in what way well trained? would just be specific.

Line 328 That is very interesting that many tested unsure of status.

Discussion

What reasons do you see for the higher rates here?

What recommendations on frequency of testing would you look for? Given high incidence, would seem testing more frequently would be needed to meet 95-95-95.

6. PLOS authors have the option to publish the peer review history of their article (what does this mean?). If published, this will include your full peer review and any attached files.

**Do you want your identity to be public for this peer review?** For information about this choice, including consent withdrawal, please see our Privacy Policy.

Reviewer #1: No

Reviewer #2: No

---

## [Decision Letter · Decision Letter 1]

1 Apr 2022

PGPH-D-21-00782R1

Determinants of testing for HIV among young people in Uganda. A nested, explanatory-sequential study

Dear Dr. Kalibbala,

Thank you for submitting revision to your manuscript to PLOS Global Public Health. The reviewers have gone through the revisions you sent but have pointed out some outstanding revisions. I have also read the manuscript and believe that it is worth considering for publication. Therefore, we invite you to submit a revised version of the manuscript that addresses the points raised during the review process. This will be the last time to make revisions and failure to address them carefully will results in rejection of the paper.

We look forward to receiving your revised manuscript.

Kind regards,

Kwabena Obeng Duedu, PhD

Academic Editor

Journal Requirements:

1. Your co-authors, Steven Kiwuwa Mpungu (mkiwuwa@yahoo.com), Bashir Ssuna (sbn144@gmail.com), Happiness Mberesero (happymberesero@gmail.com), Anne Katahoire (annekatahoire@yahoo.co.uk), Mari Armstrong-Hough (mah842@nyu.edu), 
Joan N. Kalyango (nakayaga2001@yahoo.com), and Victor Musiime (musiimev@yahoo.co.uk), have not confirmed authorship of the manuscript. We have resent them the authorship confirmation email; however please check that the above email address for them is correct and follow up personally to ensure they confirm. Please note that we cannot pass your manuscript to Production until we have received confirmations from all co-authors.  2. Please update your Competing Interests statement. If you have no competing interests to declare, please state: “The authors have declared that no competing interests exist.”  3. Please ensure that you refer to Table 4 in your text as, if accepted, production will need this reference to link the reader to the table.  4. Please review your reference list to ensure that it is complete and correct. If you have cited papers that have been retracted, please include the rationale for doing so in the manuscript text, or remove these references and replace them with relevant current references. Any changes to the reference list should be mentioned in the rebuttal letter that accompanies your revised manuscript. If you need to cite a retracted article, indicate the article’s retracted status in the References list and also include a citation and full reference for the retraction notice.

Additional Editor Comments (if provided):

Reviewers' comments:

Reviewer's Responses to Questions

**Comments to the Author**

1. If the authors have adequately addressed your comments raised in a previous round of review and you feel that this manuscript is now acceptable for publication, you may indicate that here to bypass the “Comments to the Author” section, enter your conflict of interest statement in the “Confidential to Editor” section, and submit your "Accept" recommendation.

Reviewer #1: All comments have been addressed

Reviewer #2: (No Response)

2. Does this manuscript meet PLOS Global Public Health’s publication criteria? Is the manuscript technically sound, and do the data support the conclusions? The manuscript must describe methodologically and ethically rigorous research with conclusions that are appropriately drawn based on the data presented.

Reviewer #1: Partly

Reviewer #2: Yes

3. Has the statistical analysis been performed appropriately and rigorously?

Reviewer #1: Yes

Reviewer #2: Yes

4. Have the authors made all data underlying the findings in their manuscript fully available (please refer to the Data Availability Statement at the start of the manuscript PDF file)?

Reviewer #1: Yes

Reviewer #2: Yes

5. Is the manuscript presented in an intelligible fashion and written in standard English?

Reviewer #1: Yes

Reviewer #2: Yes

6. Review Comments to the Author

Reviewer #1: All of the comments from reviwers were addressed but not fully responsed.

The TEXT needs big improvement. For exsample, two numbers in lines 99-100 were missed.

Reviewer #2: Thank you for including more info on gender.

Line 42. would rework this sentence. The prevalence testing for HIV was 80.2%

It will likely be misread, though is accurate.

Line 64-5, would state year of figure (published in 2016 and I suspect it has increased substantially since then). Would also stress than 95-95-95 requires 95% of those who are HIV+ to know their status. This isn't the same as saying 95% of youth need to be tested. depends on epidemic and risk though certainly everyone knowing their status would lead to this goal.

Table 4 is still a concern that shares more info than we need. Ethically this is a concern and I would not include this detailed info here. This is PHI and not clear the participants fully agreed to have this level of info shared. Technically, someone reading this in the area might be able to guess who someone was.

We really do not need to precise age so no reason to provide something this identifying and not sure even area they are from is needed. Could just provide percent of participants from each area, age median, range in age and breakdown in gender.

7. PLOS authors have the option to publish the peer review history of their article (what does this mean?). If published, this will include your full peer review and any attached files.

**Do you want your identity to be public for this peer review?** For information about this choice, including consent withdrawal, please see our Privacy Policy.

Reviewer #1: No

Reviewer #2: No

---

## [Decision Letter · Decision Letter 2]

17 Aug 2022

PGPH-D-21-00782R2

Determinants of testing for HIV among young people in Uganda. A nested, explanatory-sequential study

Dear Mr. Kalibbala,

Thank you for submitting your manuscript to PLOS Global Public Health. The reviewers have carefully considered the revised version of this manuscript and feel that it has merit and is worth considering for publication. However, there are some minor revisions which need addressing in order to fully meet PLOS Global Public Health’s publication criteria. Therefore, we invite you to submit a final revised version of the manuscript that addresses the points raised during the review process. Kindly note that this will be the final opportunity to adequately revise and submit the manuscript for consideration.

EDITOR: 

Kindly rephrase the first sentence in the results section of the abstract and in line 187 (page 9) relating to the rate of uptake of HIV testing services to avoid any ambiguity as has been suggested by Review #2.Review the manuscript for grammatical and editorial errors for better clarity.

We look forward to receiving your revised manuscript.

Kind regards,

Edina Amponsah-Dacosta, Ph.D., MPH

Academic Editor

Journal Requirements:

Reviewers' comments:

Reviewer's Responses to Questions

**Comments to the Author**

1. If the authors have adequately addressed your comments raised in a previous round of review and you feel that this manuscript is now acceptable for publication, you may indicate that here to bypass the “Comments to the Author” section, enter your conflict of interest statement in the “Confidential to Editor” section, and submit your "Accept" recommendation.

Reviewer #2: All comments have been addressed

Reviewer #3: (No Response)

2. Does this manuscript meet PLOS Global Public Health’s publication criteria? Is the manuscript technically sound, and do the data support the conclusions? The manuscript must describe methodologically and ethically rigorous research with conclusions that are appropriately drawn based on the data presented.

Reviewer #2: Yes

Reviewer #3: (No Response)

3. Has the statistical analysis been performed appropriately and rigorously?

Reviewer #2: Yes

Reviewer #3: (No Response)

4. Have the authors made all data underlying the findings in their manuscript fully available (please refer to the Data Availability Statement at the start of the manuscript PDF file)?

Reviewer #2: Yes

Reviewer #3: (No Response)

5. Is the manuscript presented in an intelligible fashion and written in standard English?

Reviewer #2: Yes

Reviewer #3: (No Response)

6. Review Comments to the Author

Reviewer #2: 

I would still recommend changing, for clarity, the sentence in the abstract that reads "The prevalence testing for HIV was 80.2%." This will be misread as 80% have HIV.

Reviewer #3: 

The manuscript explores factors influencing HIV testing among young persons aged 15 – 24 years in Wakiso district, Uganda. It applies a mixed methods approach to examine potential barriers to HIV testing in this age group.Overall, the work is well presented and only requires minor revisions.Minor commentsHow does HIV testing prevalence in these young people compare with older persons (>24 years)? What is the estimated testing prevalence in older folks? Wouldn’t most of the identified barriers also apply to older persons, resulting in broad policy implications?Results: Tables 3. Seems longer distance (>10kms from a facility) was associated with a decreased likelihood of testing, but language in lines 196 – 201 suggests otherwise. Could the authors restructure this?Table 2. Please provide N for each category (Testing N=xx, Not Testing N=yy).Line 235 – 236; the sentence beginning with “For example, one of …” is incompleteLine 248 – 249. Please check grammar.Line 251. Change “introduce” to “introduced”. Line 252, change “attends” to “attended”.Line 253 – 254. Sentence beginning with “Young people also …” not clear. Please reword.

7. PLOS authors have the option to publish the peer review history of their article (what does this mean?). If published, this will include your full peer review and any attached files.

**Do you want your identity to be public for this peer review?** For information about this choice, including consent withdrawal, please see our Privacy Policy.

Reviewer #2: No

Reviewer #3: **Yes: **Godfrey Bigogo

---

## [Editor Report · Decision Letter 3]

17 Oct 2022

Determinants of testing for HIV among young people in Uganda. A nested, explanatory-sequential study

PGPH-D-21-00782R3

Dear Mr. Kalibbala,

We are pleased to inform you that your manuscript 'Determinants of testing for HIV among young people in Uganda. A nested, explanatory-sequential study' has been provisionally accepted for publication in PLOS Global Public Health.

Best regards,

Edina Amponsah-Dacosta, Ph.D., MPH

Academic Editor
